# Effect of Al Concentration on Basal Texture Formation Behavior of AZ-Series Magnesium Alloys during High-Temperature Deformation

**DOI:** 10.3390/ma16062380

**Published:** 2023-03-16

**Authors:** Kibeom Kim, Yebin Ji, Kwonhoo Kim, Minsoo Park

**Affiliations:** 1Department of Marine Design Convergence Engineering, Pukyong National University, 45 Yongso-ro, Nam-gu, Busan 48513, Republic of Korea; 2Department of Metallurgical Engineering, Pukyong National University, 45 Yongso-ro, Nam-gu, Busan 48513, Republic of Korea; 3Institute of Multidisciplinary Research for Advanced Materials (IMRAM), Tohoku University, Katahira 2-1-1, Sendai 980-8577, Japan

**Keywords:** magnesium alloy, aluminum content, crystallographic texture, uniaxial compression, AZ-series magnesium alloy, stacking fault energy (SFE)

## Abstract

Magnesium and its alloys have been restricted in their industrial applications due to problems related to their formability. To overcome this issue, controlling the crystallographic texture is important, and the texture formation mechanism should be investigated in relation to factors including deformation conditions and solute atoms. In particular, the effects of solute atoms on the texture formation behavior should be further analyzed because they can considerably affect the deformation behavior. Thus, in this study, to clarify the effect of aluminum concentration on the texture formation behavior and microstructure, high-temperature uniaxial compression tests were conducted on three types of AZ-series magnesium alloys (AZ31, AZ61, and AZ91). Compression was conducted at 673 K and 723 K, with strain rates of 0.05 s^−1^ and 0.005 s^−1^, up to a true strain of −1.0. Cylindrical specimens were prepared from a rolled plate that had a (0001) basal texture and was compressed parallel to the c-axis of the grains. Consequently, work softening and fiber texture formation were observed in all the specimens. During the deformation, the development of grain boundaries, which is a typical characteristic of continuous dynamic recrystallization (CDRX), was observed, and the (0001) texture was highly developed with increasing Al content. Although each alloy was associated with the same deformation conditions and mechanisms, the AZ31 alloy exhibited a non-basal texture component. The stacking fault energy contributed to the generation of slip systems and gliding, and it was seen as the main reason for texture variation.

## 1. Introduction

Owing to their excellent properties, such as low density and high specific strength, magnesium and its alloys are potential alternatives to conventional structural materials. However, they have low formability at room temperature due to the lack of active slip systems associated with the hexagonal close-packed crystal structure [1,2,3]. To overcome this issue, the alloys require processing at high temperatures to activate more slip systems. Unfortunately, the occurrence of a basal texture induces plastic anisotropy and mechanical defects, such as fractures or gliding, during the deformation process [4,5,6]. To advance the application of magnesium and its alloys, understanding the texture formation mechanism and controlling its development became important.

Fukutomi et al. investigated the texture formation behavior of Al–Mg system alloys that have an FCC structure and different magnesium concentrations during high-temperature uniaxial compression. From the texture measurements, including X-ray diffraction (XRD) and electron backscattered diffraction (EBSD) results [7,8,9], the following phenomena were observed: (1) Deformation was accompanied mainly by dislocations, and variations in factors, such as the strain rate and deformation temperature, made the Al alloy have textures in specific orientations rather than in random directions; (2) A different magnesium concentration in the alloy affected the intensity of the texture after the deformation process; and (3) The orientations were stable and hardly changed during the high-temperature compression deformation. Similar behaviors were also observed in the results of Fe–Si alloys with BCC structures [10]. Moreover, one of the authors investigated texture formation behaviors during high-temperature uniaxial compression using AZ80 magnesium alloys to clarify the effects of deformation factors, including varying compression conditions and different initial textures [11,12]. They found that the main components of the texture and its sharpness varied depending on the deformation conditions. Through the results, it could be assumed that the formation of basal texture in magnesium alloys can be formed in a similar manner to that of other alloys that have different structures. When the development of basal texture occurred during the high temperature deformation of AZ80 alloy, the peak stress exceeded a specific point and the stress exponent neared five. The conditions such as higher strain rate or lowered temperature made the flow stress increase. In addition, the dislocation glide model became the dominant mechanism. This mechanism activated slip systems that had a higher Schmid factor along with the applied stress.

Moreover, one of authors has investigated the effect of Ca precipitation on the texture development of AZ magnesium alloys. [13] Through the results, the AZ series magnesium alloy showed that the higher Al content made them have a more-sharpened basal texture. In addition, the basal texture formation behavior was shown more sharply in AZ61 than that of AZ31 [14]. Although the conversion of deformation mechanisms from dislocation glide to solute atom drag seemed more effective in Mg alloys with a higher Al content rather than a lower one, the reported results were shown as the contrast.

Although there are many studies about the affected properties improved by the Al content [15,16] or the effect of other elements on the texture formation behaviors in Mg alloys [17,18,19,20], there are less results related to the effect of solute content on the texture formation behavior of the magnesium alloys in these results, from the view point of the interaction between the solute atoms and the dislocation glide.

Therefore, in this study, to clarify the effect of Al concentration on the texture formation behavior, AZ-series magnesium alloys with different Al concentrations (AZ31, AZ61, and AZ91) were investigated by conducting high-temperature uniaxial compression tests.

## 2. Materials and Methods

Commercial cast AZ-series magnesium alloys with different Al concentrations were used as the materials. They were obtained from the Suzhou Chuanmao Metal Material.co. The specimens were cut from the cast ingots in to 10 mm (width) × 10 mm (length) × 4 mm (thickness) sizes, and their chemical compositions were investigated using X-ray fluorescence ((XRF) XRF-1800, SHIMADZU, Tokyo, Japan). Table 1 shows the chemical compositions of the AZ-series magnesium alloys used in this study.

Rectangular plates with the dimensions of 60 mm (width) × 60 mm (length) × 40 mm (thickness) were machined from ingots. These plates were hot rolled at a temperature of 673 K, with a rolling reduction of approximately 30%. For uniaxial compression tests, cylindrical specimens with a diameter of 12 mm and a height of 18 mm were cut from the rolled plates using a precision low-rate cutter (Minitom, Struers, Copenhagen, Denmark) and a milling machine (SR-10J, Star Micronics, Shizuoka, Japan). The compression direction of the specimen was set as the normal direction (ND) of the rolled plate, while the elongation directions were parallel to the transverse (TD) and rolling (RD) directions. All the specimens were annealed at 723 K for 3600 s to homogenize the microstructure and for stress relaxation. Thereafter, they were oil quenched to prevent the post-annealing effect on the microstructure.

Figure 1a–c show the microstructure and (0001) pole figures of the as-received specimens of the AZ31, AZ61, and AZ91 alloys. The compression axis is shown next to the right side. All the grains are colored in accordance with the standard triangle color code. However, the colors show the orientation of the compressed plane, while the microstructures show the transverse section of the compressed specimen. The pole densities were projected onto the compression plane. The mean pole density was used as a unit for drawing the contour lines. As shown in Figure 1, all the AZ-series magnesium alloys exhibit a basal texture, owing to rolling. The maximum pole densities of AZ31, AZ61, and AZ91 were 4.5, 3.7, and 4.0 times the mean pole densities each, and they were largely similar. Moreover, the average grain sizes of AZ31, AZ61, and AZ91 before deformation were 17.6 ± 5.0, 23.2 ± 5.0, and 39.0 ± 5.0 μm, respectively.

Uniaxial compression tests were performed at two different temperatures, namely, 673 K and 723 K, under strain rates of 5.0 × 10^−2^ s^−1^ and 5.0 × 10^−3^ s^−1^, up to a true strain of –1.0. To prevent microstructural changes after high-temperature deformation, the specimens were immediately quenched in oil after deformation.

To analyze the microstructure and texture, pre-treatments were conducted. The deformed specimens were cut parallel to the compression direction, grinded with sand paper, and polished with OP-S silica colloids with 1 µm Si–C particles. Electropolishing was conducted under a current density of 0.20 A/cm^2^ by controlling the voltage at 243 K, in a solution containing 3% perchloric acid and high-purity ethanol. The microstructure was observed using a field-emission scanning electron microscope (JSM 6360, JEOL, Tokyo, Japan) and EBSD (Hikari, AMETEK, Berwyn, PA, USA). The texture was measured using XRD (D/max-2000, Rigaku, Tokyo, Japan). The five imperfect pole figures of {0001}, {10$\bar{1}$0}, {10$\bar{1}$1}, {10$\bar{1}$2}, and {11$\bar{2}$0} were measured using the Schulz reflection method with Cu-Kα radiation. Based on the five pole figures, the orientation distribution function (ODF) was calculated using the Dahms and Bunge method [21]. Thereafter, the perfect pole figure was recalculated, and inverse pole figures (IPFs) were derived from the ODF. Each IPF texture was shown with the contour on a standard triangle, and their component was indicated using the (α, β) coordinate system.

Figure 2 shows the method used in this study to express the texture component in the coordinate system. α is the angle between the (0001) and {11$\bar{2}$0} planes, and β is the angle between the {10$\bar{1}$0} and {11$\bar{2}$0} planes. Therefore, α can reach a maximum of up to 90°, whereas β can take values of up to 30°.

## 3. Results

### 3.1. True Stress–Strain Curves

Figure 3 shows the true stress–strain curves of the AZ-series magnesium alloys deformed at temperatures of 673 K and 723 K under a strain rate of 5.0 × 10^−2^ s^−1^, up to a strain of −1.0. The true strain in the figure is given in terms of the absolute value. All the true stress–strain curves show maximum stresses in the initial stage of deformation that decrease thereafter. Namely, the curves characteristic of work softening can be seen. Furthermore, this phenomenon was observed under all the conditions tested in this study, though there are some differences in the magnitude. There was a difference in the peak stresses of AZ31 and AZ61 in the initial stages; nevertheless, the flow stresses were largely similar during the deformation process. On the contrary, the stress of AZ91 was more dynamic. AZ91 exhibited the most significant work softening at low temperatures and the lowest flow stress at high temperatures.

### 3.2. Deformation Behavior of Microstructure

Figure 4 shows the microstructural observations made by EBSD measurements at the longitudinal section of the AZ-series magnesium alloys, deformed at a temperature of 673 K under a strain rate of 5.0 × 10^−2^ s^−1^, up to a strain of −1.0. The microstructures are expressed in the same method that of Figure 1. After the deformation, the grains were almost equiaxial, rather than elongated toward the transverse direction. In addition, the microstructure showing grain orientation became more reddish during the compression process, which meant that the grains were more strongly aligned to the (0001) orientation.

Figure 5 shows the relationship between the peak stress in the true stress–strain curves and the grain size after deformation of the AZ-series magnesium alloys. As described in the previous section, the peak stress was higher at higher temperatures or lower strain rates. The grain size after deformation shows a one-to-one correspondence between the peak stresses, which varied depending on the deformation condition. The grain size after deformation decreased with an increase in the peak stress. These results suggest the occurrence of dynamic recrystallization (DRX) during the high-temperature deformation, and this process affected the texture formation behavior in the three types of AZ-series magnesium alloys.

### 3.3. Texture Development

Figure 6a–c show the (0001) pole figures for AZ31, AZ61, and AZ91, deformed at a temperature of 723 K under a strain rate of 5.0 × 10^−2^ s^−1^, up to a true strain of −1.0. The pole densities are projected using the form shown in Figure 1. Regardless of whether the deformation was applied, the distribution of the pole densities was a concentric circular pattern. This meant that the fiber texture formed, owing to hot rolling, was maintained until the final strain. The texture intensity was the only factor that varied during the compression. There was more accumulation of the (0001) texture with an increasing aluminum concentration. Those of the AZ61 and AZ91 alloys, which had a comparatively high Al content, increased more than three times, while the AZ31 alloy had an approximately two-fold increase. Moreover, the concentric circular of the pole figure tilted toward a specific direction in AZ31.

Figure 7 shows the (0001) pole densities of the AZ-series magnesium alloys deformed at 723 K under a strain rate of 5.0 × 10^−2^ s^−1^ with the increase in the strain. The (0001) pole densities gradually increased with an increase in the true strain of the three types of magnesium alloy. Because the intensity of the AZ91 alloy increased slightly, it was lower than those of the others during the middle stage of the deformation. However, the intensity of the AZ91 alloy was higher than those of the AZ61 and AZ31 alloys after the final deformation. Therefore, it was implied that the Al content affected the development of the (0001) pole density, and its increment resulted in a higher intensity of the (0001) poles.

Figure 8 shows the relationship between the maximum-axis density from the inverse pole figure and the peak stress from the true stress–strain curves. AZ61 and AZ91 magnesium alloys showed a basal texture component, regardless of the peak stress. Moreover, the main component of the texture was (0001) when the peak stresses from the stress–strain curves were greater than 15–20 MPa under high-temperature uniaxial compression. Moreover, the intensities of the (0001) texture increased with an increase in the peak stress. However, the texture components of AZ31 varied depending on the deformation condition. Although the AZ31 specimens were compressed under the same condition as that of AZ61 or AZ91, non-basal texture components, such as (10,0) or (18,30), were developed except under a stress of over 30 MPa.

## 4. Discussion

### 4.1. Deformation Mechanism

From the results shown in Section 3, the deformation made the AZ-series alloy undergo DRX, and the basal texture was strengthened during the deformation. Prior to clarifying the effects of the Al content and mechanism of the basal texture formation, the deformation condition should be checked.

According to the von Mises theory, magnesium alloys require the activation of at least five or more independent slip systems to accompany the deformation [13]. However, magnesium alloys are generally deformed by basal slip systems, and the lack of slip systems becomes the major problem of formability. To compensate for this, tensile twinning should be activated to provide more deformation systems when the deformation process is performed at a comparatively low temperature. However, the occurrence of twinning is inhibited with the increase in the temperature, and non-basal slip systems become more activated [22,23,24]. Although there are some reports that twinning systems were observed even under high-temperature deformation, it was difficult to attribute twinning as the major deformation mechanism in this study.

The strain rate and temperature, as deformation parameters, also seemed to contribute to the deformation behavior. In the case of the AZ80 alloy, these parameters affected the flow stress during compression [11]. In particular, the flow stress decreased when deformation was conducted under lowered strain rate or high-temperature conditions. A boundary where the texture component was altered after the deformation was determined through the stress exponent. When the stress exponent was near five, the dominant mechanism of the dislocation glide was in free flight. During the glide mechanism, dislocations that activated first were the ones that had a higher Schmidt factor. Otherwise, when the stress exponent was approximately three, the solute atom drag affected the deformation behavior. In this case, the activated slip systems changed because the dislocations were dragged by the solute atoms, the so-called Cottrell atmosphere. However, continuous DRX behaviors were observed regardless of the stress exponent value after deformation. Similarly, the basal texture formed prior to the deformation of the AZ-series alloy was maintained in most cases with higher stresses. On the other hand, a non-basal texture component developed in AZ31 even under comparatively high stress. Considering that the solute atom drag mechanism was highly activated with the increase in the solute atoms and that alloys with a higher Al content maintained the development of the basal texture, the tilted orientation might have occurred due to other reasons. Moreover, because of the geometrical alignment of the HCP structure before the compression, the slip systems, including in the <a> direction, could not be easily activated. It was rational to consider the activation of the pyramidal II <c+a> slip systems.

Mg_17_Al_12_ precipitation mainly occurs in the Mg–Al binary phase, and it contributes to the strength of magnesium alloys. However, according to its phase diagram, the Mg_17_Al_12_ phase in the AZ91 alloy is decomposed at a temperature of 643 K, and its decomposition temperature decreases as the composition reaches that of pure Mg [25]. Moreover, the thermal properties of precipitation phase such as Mg_17_Al_12_ have been already investigated in our previous study [13,26]. In particular, the results were obtained through the differential scanning calorimetry (DSC) method, using the AZ61 alloy and AZX611 alloy. In the DSC cooling and heating curves of the AZ61 alloy, it was observed that the mg-matrix solidified around 883 K (620 °C) and melted at 873 K (610 °C). The initial and end temperatures, according to the reaction, were different to each other, but peak temperatures were similarly observed in 878 K (615 °C). Moreover, in the second phase, the Mg_17_Al_12_ phase in the cast, ingot was observed by transmission electron microanalysis (TEM). They were discomposed or precipitated around 688 K (415 °C). Similarly, AZX611 alloy showed two kinds of different precipitation, which are Mg_17_Al_12_ and Al_2_Ca. Because the Ca component was not soluted into the matrix, the matrix of AZX611 alloy showed a more lowered Al content than the AZ61 alloy. However, even in AZX611, the peak temperature of the Mg_17_Al_12_ phase occurred in same point. Therefore, only the effect of the Al content was considered as the main reason for the variation in the texture intensity. Because the Al content soluted into the matrix at temperatures of 673 K and 723 K, they had to be conjectured as solute atoms, which could affect the lattice distortion on stacking fault energy and axis ratio. Although Zinc can still affect the formation of different phases, it is difficult to conceive that it significantly affected the overall deformation behavior under low concentrations compared to Al.

Therefore, in this study, the gliding of the pyramidal II <c+a> slip systems and the DRX process were considered the major deformation modes, and the Al content could affect the deformation behavior.

### 4.2. Mechanism of Basal Texture Formation by DRX

The following observations can be made from the results of the high-temperature uniaxial compression tests conducted on AZ-series magnesium alloys: (1) The major deformation mechanism is the glide of pyramidal II <c+a> and the occurrence of DRX; (2) Increasing Al concentration contributed to the strengthening of the basal texture; (3) A tilted texture from the basal pole in AZ31 may be developed due to the other reasons, rather than the variation in deformation mechanism. Based on these results, the mechanism of basal texture formation is discussed.

Figure 9a–c show the grain structure maps at the mid-plane section observed using EBSD measurements of the AZ61 and AZ91 alloys deformed at 723 K under a strain rate of 5.0 × 10^−2^ s^−1^ up to strains of (a) −0.4, (b) −0.7, and (c) −1.0. The red color grains are within 15° from the (0001)-oriented grain, according to the standard triangle. Moreover, the misorientation angle, which ranges from 15° to 180°, is indicated by a black line: high-angle grain boundary (HAGB). The fraction of the (0001)-oriented grain gradually increases when the true strain increases up to −1.0, while the average grain size decreases gradually.

Table 2 presents the fractions of the LAGB and HAGB of the grain structure maps shown in Figure 9. Misorientation angles exceeding 15° correspond to HAGBs, while those lower than 15° correspond to LAGBs. Moreover, they were divided into two parts with the misorientation angle ranging from 2° to 5° and from 5° to 15°. In the results, the fraction of LAGBs ranging from 2° to 5° decreased with the increment in the strain, while that of HAGBs increased gradually. This behavior implies that the HAGBs were developed from the small misorientation via the formation of LAGBs by the accumulation and consuming of dislocation during the deformation. This is typical of CDRX, and it refined the grains while increasing the fraction of the (0001)-oriented grain [27,28].

Fukutomi et al. suggested that the texture orientation in the Al–Mg alloy formed after deformation is associated with a low Taylor factor [9]. The Taylor factor is determined by the number of slip systems contributing to the deformation. Therefore, a higher Taylor factor means that a higher number of slip systems engaging in the deformation can be applied to the grain. In other words, the stored energy in each grain increases. In addition, the Taylor factor for the grain was calculated by its orientation. This difference in the Taylor factor on each grain rolled is a factor determining which grains would consume others or would be diminished. From the many results on the factors affecting dislocation movement, such as the strain rate in the AZ31 alloy [29] or precipitation in the AZX611 [13] magnesium alloy, a similar phenomenon was observed.

As mentioned in Section 4.1, because the six pyramidal II <c+a> slip systems are activated in the grains having (0001) orientation, this orientation exhibited the lowest Taylor factor. Therefore, the development of the basal texture, shown in this study, can be elucidated as follows: the other orientation would fade when the stress is applied along the c-axis, by the growth of the (0001) grains. Based on this, the development of the non-basal texture component in AZ31 seemed to be the result of the Taylor factor variation. The Taylor factor near the (0001) zone increased higher than the orientation observed in the AZ31 specimen. This meant that the number of activated slip systems was higher than that in AZ61 or AZ91, as a result of the difference in the Al content in the Mg alloy.

### 4.3. Effect of Al Content on Deformation

Figure 10 shows the relationship between the aluminum concentration and the (0001) pole density deformed up to a strain of −1.0 at different temperatures and strain rates. The results show that the texture intensity increases when compared to the initial value, while maintaining the major component as (0001), and the strengthening was higher with the increase in the aluminum concentration. Evidently, the aluminum concentration contributed to the formation and accumulation of a sharp (0001) texture.

From the discussion in Section 4.1, it was considered that the Al content in the AZ alloy affected the deformation behavior by a solid solution, rather than precipitation. In addition, because the dislocation glide was the deformation mechanism in this study, the effect of Al concentration was conjectured as a lattice distortion.

Many researchers tried to investigate the stacking fault energy of magnesium alloys by calculations or experiments [18,29,30]. In particular, Somekawa et al. calculated the stacking fault energy with respect to the aluminum concentration in AZ-series magnesium alloys; Table 3 presents the results. The SFE of AZ31 was approximately 5 times higher than those in AZ91, and the SFE decreased significantly when the aluminum concentration increased in the magnesium alloy.

Pure magnesium and aluminum are known as high-SFE alloys [2]. A high SFE means that the alloy has a low equilibrium separation distance. Because this value shows a balanced distance between two divided partial dislocations, it also means that cross slip occurs more easily. On the contrary, as the SFE decreases, perfect dislocation becomes dominant rather than being partial. In addition, as mentioned in the previous section, the development of the basal texture is mainly the result of the lowest value of the Taylor factor in the (0001) orientation. Specifically, this value corresponds to the six perfect-pyramidal II slip systems. Because the cross-slip systems, owing to the SFE, provide dislocation paths, their activation seems to contribute to the increase in the Taylor factor [9]. This implies that a non-basal texture component can develop in AZ31, the component most similar to that of the pure state.

In this study, the magnesium alloy containing a higher Al content showed a greater increase in the (0001) intensities, whereas AZ31 showed the development of a tilted texture from the basal axis. These results well support the assumption regarding the relationship between the SFE and the Al content in the Mg alloy. Consequently, the Al content in the AZ-series magnesium alloys contributed to lower SFE and made the perfect slip predominant, inhibiting cross slip. This effect made the magnesium alloy have a stronger (0001) texture intensity.

These results showed that the basal texture will be formed or developed when the (0001) orientation had the lowest Taylor factor during the deformation. In addition, this orientation was set in the most of cases. Considering that the value of Taylor factor was determined by the combination of slip systems contributing to the deformation, non-basal texture orientations could be developed by the variation in the combination. In this viewpoint, additive elements to magnesium alloy, which could enhance the SFE, seem to be the key to improve formability. However, further studies about this assumption are required.

## 5. Conclusions

To clarify the effect of aluminum concentration on texture formation behavior and microstructure, three types of AZ-series magnesium alloys with different aluminum concentrations (AZ31, AZ61, and AZ91) were experimentally studied by conducting high-temperature uniaxial compression tests. The main results of the study are as follows:Continuous types of dynamic recrystallization were observed in all deformation conditions. The microstructures varied through the formation of LAGBs and the development of HAGBs.A fiber texture was formed by recrystallization. The grains which would be grown up or consumed were determined according to the orientation they had before the deformation.The (0001) orientation had the lowest Taylor factor than the other orientations because the slip systems contributing to deformation on this orientation were the combination of the pyramidal <c+a> slip systems. This made the basal texture to be strengthened.The Al concentration in the magnesium alloy contributed to a decrease in the stacking fault energy and inhibited cross slip, leading to the strengthening of the basal texture. On the contrary, the variation in SFE, which can affect the Taylor factor, has possibilities to inhibit the development of basal textures in magnesium alloys.

## Figures and Tables

**Figure 1 materials-16-02380-f001:**
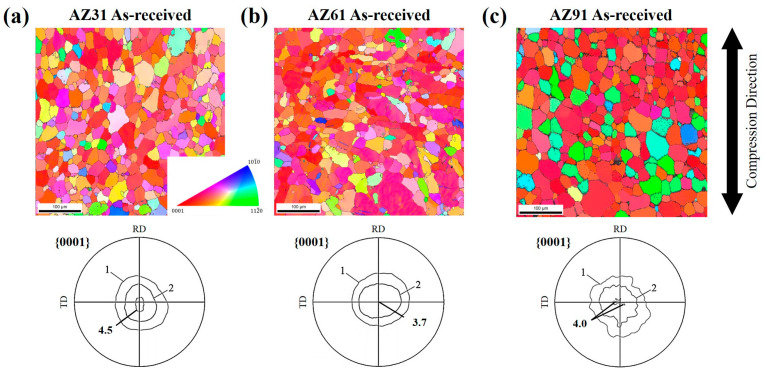
The microstructures and (0001) pole figures of three types of specimens before deformation: (**a**–**c**) AZ31, AZ61, and AZ91. The pole densities are projected onto the compression plane, and the mean density is used as a unit.

**Figure 2 materials-16-02380-f002:**
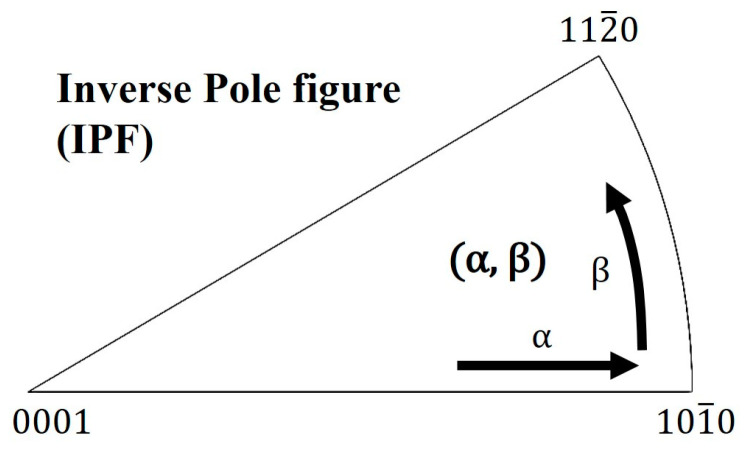
Inverse pole figure triangle method of expressing the textures. The main texture component is mainly observed as a contour plot, and the highest point is represented in the (α, β) coordinate system.

**Figure 3 materials-16-02380-f003:**
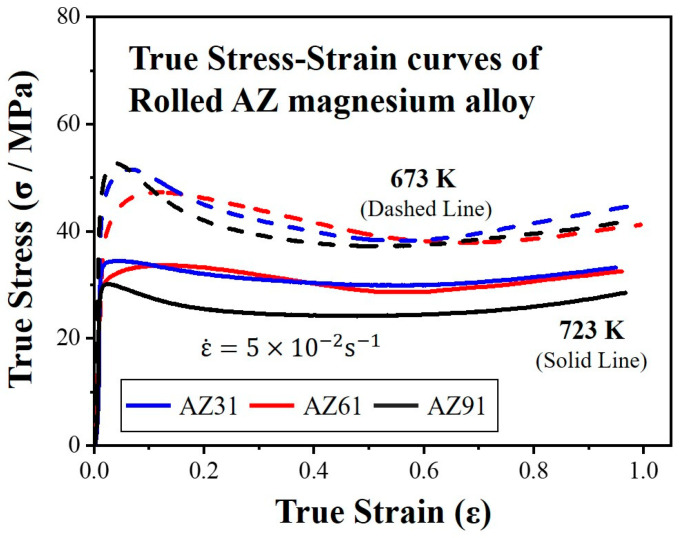
True stress–true strain curves of the AZ-series magnesium alloy with three different solute concentrations deformed at temperatures of 673 K and 723 K under a strain rate of 5.0 × 10^−2^ s^−1^.

**Figure 4 materials-16-02380-f004:**

Grain structure map observed at the mid-plane section by EBSD measurement after deformation at a temperature of 673 K up to a strain of −1.0 under a strain rate of 5.0 × 10^−2^ s^−1^. (**a**–**c**) AZ31, AZ61, and AZ91.

**Figure 5 materials-16-02380-f005:**
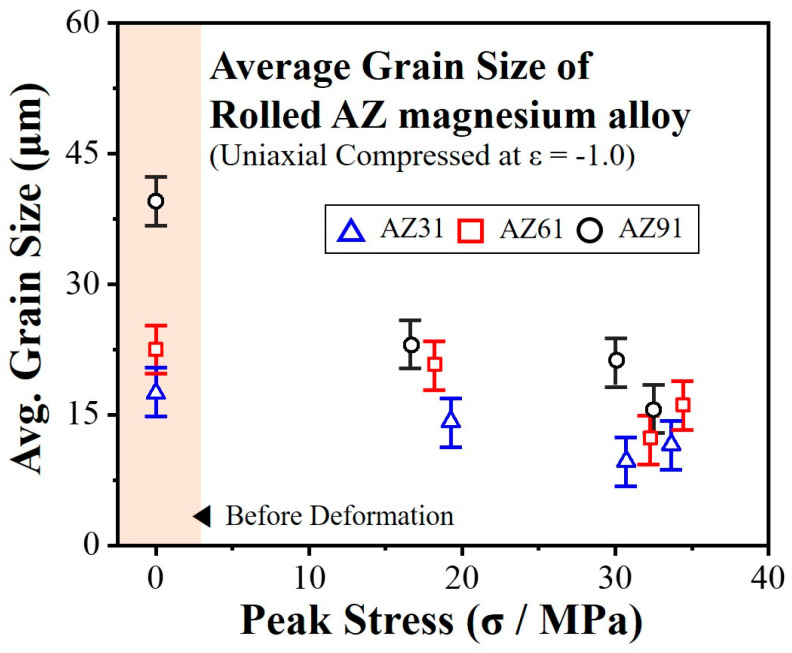
Relationship between the grain size after uniaxial compression and the peak stress.

**Figure 6 materials-16-02380-f006:**
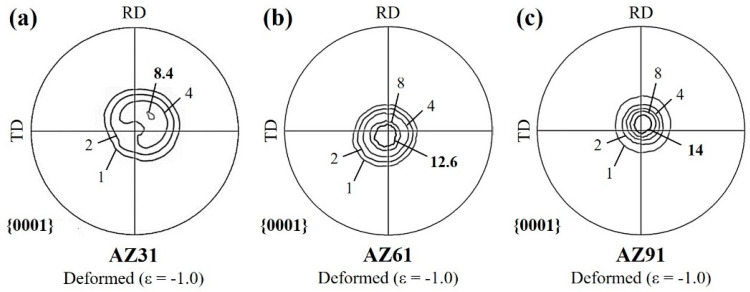
(0001) pole figures of (**a**) AZ31, (**b**) AZ61, and (**c**) AZ91 alloys deformed at a temperature of 723 K under a final strain rate of 5.0 × 10^−2^ s^−1^ up to a strain of −1.0.

**Figure 7 materials-16-02380-f007:**
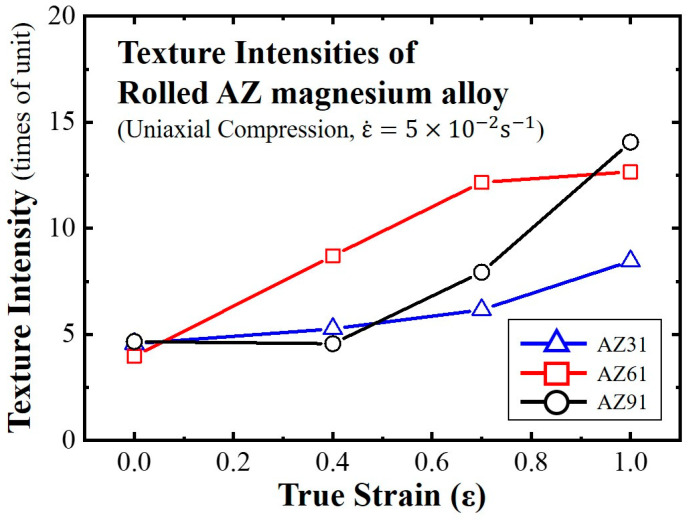
Relationship between the {0001} pole density after uniaxial compression and true strain.

**Figure 8 materials-16-02380-f008:**
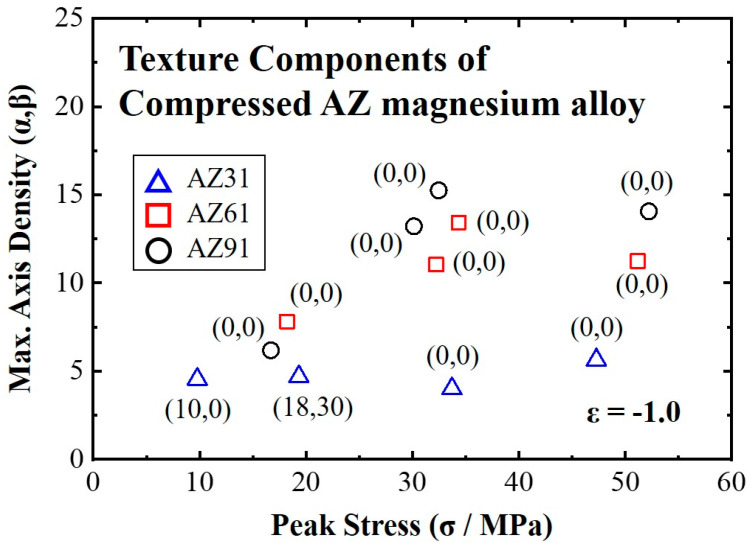
The components and intensities of deformed texture according to the peak stress.

**Figure 9 materials-16-02380-f009:**
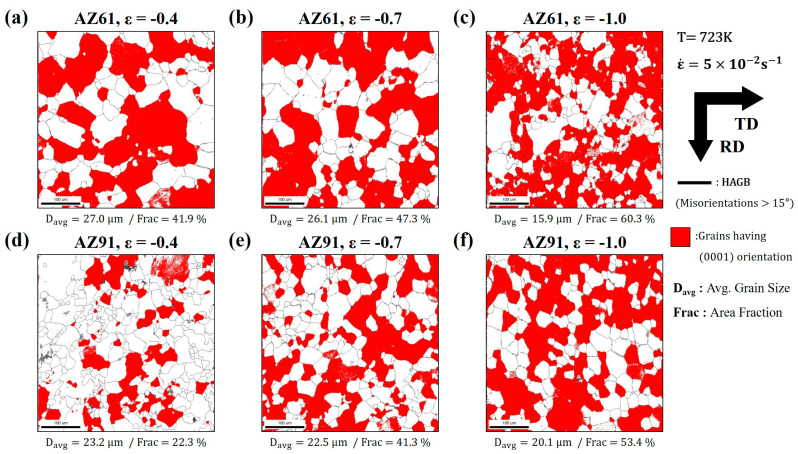
Grain structure observed at mid-plane sections after deformation at 723 K with a strain rate of 5.0 × 10^−2^ s^−1^ up to true strains of (**a**,**d**) −0.4, (**b**,**e**) −0.7 and (**c**,**f**) −1.0. ((**a**–**c**): AZ61, (**d**–**f**): AZ91) Grains having a (0001) orientation are colored in red. The average grain size and the fraction of the (0001) grains are shown in each map.

**Figure 10 materials-16-02380-f010:**
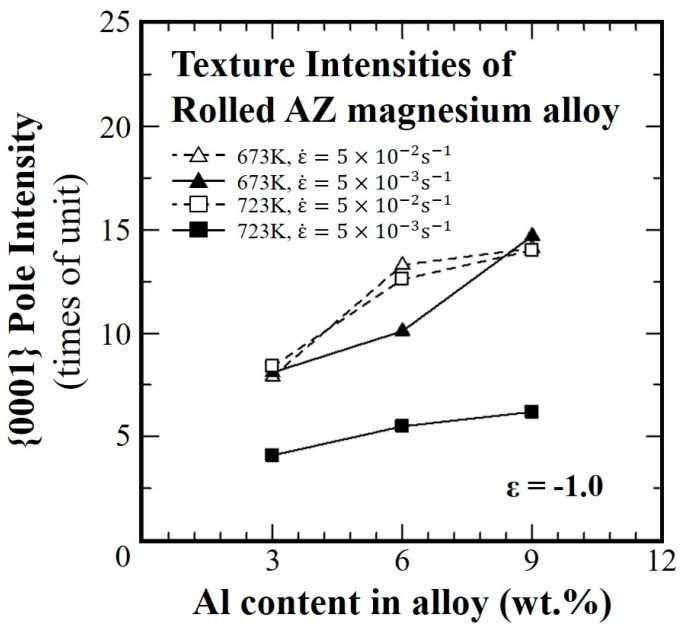
Aluminum concentration dependence of the pole densities at temperatures of 673 K and 723 K under strain rates of 5.0 × 10^−2^ s^−1^ and 5.0 × 10^−3^ s^−1^ up to a strain of −1.0.

**Table 1 materials-16-02380-t001:** Chemical composition of AZ-series magnesium alloy (wt.%).

Alloy Code	Mg	Al	Zn	Mn
AZ31	Bal.	3.12	0.79	0.19
AZ61	Bal.	6.20	0.53	0.15
AZ91	Bal.	9.08	0.59	0.15

**Table 2 materials-16-02380-t002:** Fraction of the boundaries with different misorientations based on the microstructure measured by EBSD of AZ61 and AZ91 alloys deformed at 723 K under a strain rate of 5.0 × 10^−2^ s^−1^.

AZ61	LAGB Fraction (%)		HAGB Fraction (%)
Strain	2–5°	5–15°	15–180°
−0.4	16.4	11.7	71.9
−0.7	13.5	12.4	74.1
−1.0	6.2	12.6	81.2
AZ91	LAGB Fraction (%)		HAGB Fraction (%)
Strain	2–5°	5–15°	15–180°
−0.4	16.4	11.7	71.9
−0.7	13.5	12.4	74.1
−1.0	6.2	12.6	81.2

**Table 3 materials-16-02380-t003:** Stacking fault energies of pure magnesium and its alloys (calculated) [30].

Materials	Pure Mg	AZ31	AZ61	AZ91
Stacking fault energy (mJ/m^2^)	78.0	27.8	16.4	5.8

## Data Availability

The datasets generated during and/or analyzed during the current study are available from the corresponding author on reasonable request.

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
