# Peer review of "Effect of Al Concentration on Basal Texture Formation Behavior of AZ-Series Magnesium Alloys during High-Temperature Deformation"

_materials, 2023, doi:10.3390/ma16062380_

Round 1
Reviewer 1 Report
1. “Dislocation free flight” appears several times in the manuscript, such as on Page 2 line 64 and
Page 11 line 347. What’s the "dislocation free flight"?
2. Grammatical errors should be corrected e.g. on page 2 line 66 there is: "Schmidt factor" but should be “Schmid factor”, on page 8 line 266 there is: "Mg17Al12…" but should be " Mg17Al12…", etc.
3. In the introduction, it should more strongly be articulated what is a scientific novelty in this manuscript. As well as a too general review of references, therefore the introduction needs improvement. A review in the introduction of only 13 references is not enough, so many relevant published papers should be cited and discussed, please read the following article. Without it, it is more of an engineering problem solution than a scientific manuscript. The literature review should be strengthened, please read the following articles.
1) Huang, W., Chen, J., Zhang, R., Yang, X., Jiang, L., Xiao, Z., & Liu, Y. (2022). Effect of deformation modes on continuous dynamic recrystallization of extruded AZ31 Mg alloy. Journal of Alloys and Compounds, 897, 163086.
2) Xu, X., Hou, H., Zhao, Y., & Liu, F. (2017). Nonequilibrium solidification, grain refinements, and recrystallization of deeply undercooled Ni-20 At. Pct Cu alloys: effects of remelting and stress. Metallurgical and Materials Transactions A, 48, 4777-4785.
3) Zhang, D., Liu, C., Jiang, S., Gao, Y., Wan, Y., & Chen, Z. (2023). Effects of dynamic recrystallization mechanisms on texture evolution in Mg-Gd-Y-Zr-Ag alloy during hot compression. Journal of Alloys and Compounds, 169190.
4. Page 2: The authors study the effect of Al concentration on basal texture formation behavior of AZ-series magnesium alloys during high-temperature deformation. According to Table 1, the chemical composition of Zn content as well as Mn content is somewhat different, especially the Zn content is significantly different (up to 0.26 wt.%), whether the differences in Zn or Mn content different have an impact on the results?
5. Page 3 line 110: "The average grain sizes before deformation for AZ31, AZ61, and AZ91 are 17.6, 23.2, and 39.0 μm, respectively." It seems that no specific evidence, such as OM or EBSD photos, is given in the manuscript.
6. Page 5 line 167: The authors state that "the microstructure showing grain orientation became more reddish during the compression process, which means that the grains were more strongly aligned to the (0001) orientation." There seems to be a lack of EBSD graphics for comparison.
7. Page 8 line 236: "To compensate for this, tensile twinning is activated to provide more basal slip systems when the deformation is performed at a comparatively low temperature. " Why does the activation of the tensile twinning provide more basal slip systems?
Author Response
First, the authors would like to appreciate on you for taking your valuable time to consider and evaluate our submission carefully. We assure that these reviews provided a good opportunity to reconsider various factors that were overlooked, and to improve the quality of the work.
In this letter, the answers to the comments received from each reviewer were prepared in a table. All the comments and answers for the revision would be shown in next page. Each order of questions is the same with the order presented in the revision history, and the answers are written below them. Author's comments on reviews are in black, and actual revisions are shown in blue. Although the completed file was also attached to reply, but it should be thought that the finalized file reflects the other reviewer’s comment in same time.

Reviewer 2 Report
Notes on the article of Kibeom Kim, Yebin Ji, Kwonhoo Kim and Minsoo Park «Effect of Al concentration on basal texture formation behavior of AZ-series magnesium alloys during high-temperature deformation»
The paper reports about the effect of aluminum concentration on the texture formation behavior and microstructure of uniaxial compressed AZ31, AZ61, and AZ91 alloys. The authors showed that the compression promotes the dynamic recrystallization, the development of HAGBs and the fiber texture formation. In addition, the authors showed that the Al concentration in the magnesium alloy contributed to a decrease in the stacking fault energy and inhibited cross slip. The article has a big theoretical importance. It is an interesting and well-written report, which should be published after revisions that are listed below:
1. P. 3, L. 110. The measurement error should be given for grain size values.
2. P. 3, L. 122. It should be written «0.20 A/cm2» instead of «0.20 A/cm2».
3. Figure 5. The Y axis should be better named as «Avg. Grain Size» instead of «Avg. Grain Diameter».
4. P. 8, L. 266. It should be written «Mg17Al12" instead of "Mg17Al12».
5. It is incorrect to rely only on the phase diagram in the matter of Mg17Al12 phase particles precipitation. Are the authors sure that dynamic aging did not occurred during deformation? Did the cooling speed inhibit the SSSS decomposition? X-ray phase analysis can provide more reliable information relating to 2 questions: (1) is there the phase particles and (2) what is the concentration of aluminum in solid solution? Is it possible to form accumulations of aluminum along the grain boundaries?
Author Response

(The authors gave the same response as above.)

Round 2
Reviewer 1 Report
I think, we may proceed with publishing this paper.
Author Response
Thanks for kindly reivewing of the manuscript
Reviewer 2 Report
The authors carefully revised the text and eliminated the comments. There is only a couple of remarks left.
1. Figure 5. The Y axis are still named as «Avg. Grain Diameter» instead of «Avg. Grain Size»ÑŽ
2. It would be better if the answer to the question 2-5 was added to the text of the article.
Author Response

(The authors gave the same response as above.)
